# A Case of Concurrent Molybdenosis, Secondary Copper, Cobalt and Selenium Deficiency in a Small Sheep Herd in Northern Germany

**DOI:** 10.3390/ani11071864

**Published:** 2021-06-23

**Authors:** Carina Helmer, Regina Hannemann, Esther Humann-Ziehank, Sven Kleinschmidt, Mareike Koelln, Josef Kamphues, Martin Ganter

**Affiliations:** 1Clinic for Swine and Small Ruminants, University of Veterinary Medicine Hannover, Foundation, Bischofsholer Damm 15, 30173 Hannover, Germany; helmer@anicon.eu (C.H.); mail@tieraerztin-hannemann.de (R.H.); mail@labvetcon.de (E.H.-Z.); 2Lower Saxony State Office for Consumer Protection and Food Safety, Food- and Veterinary Institute Braunschweig/Hannover, Eintrachtweg 17, 30173 Hannover, Germany; sven.kleinschmidt@laves.niedersachsen.de; 3Institute of Animal Nutrition, University of Veterinary Medicine Hannover, Foundation, Bischofsholer Damm 15, 30173 Hannover, Germany; dr.mareike.koelln@bosch-tiernahrung.de (M.K.); josef.kamhues@tiho-hannover.de (J.K.)

**Keywords:** cobalt, copper, selenium, deficiency, molybdenosis, sheep

## Abstract

**Simple Summary:**

Mineral deficiencies are very widespread among small ruminant herds throughout the world and play a big role in small ruminant herd health management. This paper describes the occurrence of a primary molybdenosis causing secondary copper deficiency, combined with ovine white liver disease (cobalt deficiency) and white muscle disease (selenium deficiency) in a group of pastured ram lambs in Northern Germany. To the authors’ knowledge, this is the first proven report of multiple trace element deficiencies in sheep in Germany for decades.

**Abstract:**

To the author’s knowledge this paper describes the first proven report of a combined primary molybdenosis, secondary copper (Cu) deficiency, Ovine White Liver Disease—Cobalt (Co) deficiency, and selenium (Se) deficiency in a small pedigree herd of White Horned Heath sheep in Germany (8 ewes, 2 rams, 3 yearling ewes, 17 lambs) for decades. Clinical signs associated with these mineral deficiencies in a group of pastured ram lambs included emaciation, conjunctivitis, anaemia, growth retardation, discolouration of the wool and photodermatitis. Morbidities and mortalities arose in 4–6-month-old lambs despite intensive veterinary treatment in the summer of 2014 and 2015 (*n* = 13, 23% died). Se (3/5), Cu (4/7), and Co (3/3) deficiencies in combination with elevated values for Molybdenum (Mo, 2/2) were found. Hamburg is a large industrial city and an input of heavy metals from surrounding industries and coal-fired power stations in combination with a sandy, non-fertilised soil and monoculture grass species might offer a potential explanation for the severity of mineral deficiencies observed in this herd.

## 1. Introduction

Molybdenum (Mo) is an essential trace element and part of the so-called molybdenum cofactor complex, which is required for three mammalian enzymes: xanthine oxidase, aldehyde oxidase and sulphite oxidase [1]. Dietary Mo, iron (Fe) and Sulphur (S) concentrations are important factors influencing the absorption and availability of copper (Cu) [2,3]. Allen and Gawthrone [4] identified that increased Mo concentrations in forage led to interactions between molybdate and sulphide to form thiomolybdates in the rumen, which can bind Cu to form nutritionally unavailable complexes. Therefore, high levels of Mo have been associated with secondary Cu deficiency. Former mining areas are potential risk areas for the occurrence of high Mo concentrations in soil and grass [5]. Forage with moderate Mo concentrations of 5–10 mg/kg dry matter (DM) may affect bone development in young cattle (focal widenings of the growth plate, uncalcified cartilage), while concentrations greater than 10 mg/kg DM may result in diarrhoea, emaciation and achromotrichia (loss of hair pigment) [6]. However, the toxic level of Mo in forage consumed by ruminants is not well defined due to several factors. Laiblin and Stöber [7] suggest that molybdenosis might possibly occur at Mo concentrations of 1–10 mg/kg fresh matter in pasture grass if the Cu concentration is <5 mg/kg fresh matter, whilst, O’Connor et al. [8] suggested that plant Mo concentrations of 10 mg/kg DM may be a very conservative estimate for the threshold at which toxic effects may be observed. Miltimore and Mason [9] suggest that Cu:Mo ratios above 2 were suitable but lower ratios were associated with signs of Cu deficiency. The National Research Council [10] insists that Cu:Mo ratios of 10:1 are necessary for good ruminant health. Mills and Davis [11] revealed that severe diarrhoea might be a direct effect of Mo toxicity. However, most clinical effects of high Mo intake appear to be associated with the induced secondary Cu deficiency, called molybdenosis. This condition occurs under natural grazing conditions in many different parts of the world including Canada [1], the USA [12], the UK [13] and Sweden [14]. As indicated, thiomolybdates appear to induce Cu deficiency in several different ways [1]. Clinical manifestations of Cu deficiency include relatively unspecific signs such as ill thrift, emaciation and achromotrichia (cattle) or wool changes (sheep). Others include cardiovascular disorders (cattle), anaemia, growth retardation, infertility [15] and bone fragility [16]. A specific condition in sheep and goats related to Cu deficiency of pregnant ewes and young lambs results in enzootic ataxia of lambs. Most often ataxia is apparent immediately after birth, but it may also be delayed several weeks. Signs of ataxia include muscular incoordination, partial paralysis of the hindquarters, and degeneration of the myelin sheath of the nerve fibres. Lambs may be born weak or die due to the inability to suckle [15,16,17]. In addition to Cu deficiency, sheep are very susceptible to Cu toxicosis as well. Cu concentrations in pastures vary, however, most cases were caused by unintended exposure of sheep to Cu overload, induced by feeding concentrates or minerals produced for other farm animals than sheep, which provide higher Cu contents [18]. Unlike other species, sheep seem to have a limited capacity to accumulate large amounts of Cu-metallothionein in the liver, and saturation occurs very quickly. The factors precipitating clinical chronic copper toxicity are unclear, but mostly include stress, acute infections and poor nutrition [19]. Noteworthy, the decrease of liver Cu concentration after exposure to Cu overload is very long with a half-life of liver Cu concentration >2 years as demonstrated in experimentally induced chronic Cu poisoning [20]. According to EU legislation (Annex/2010/349/EU), the maximum concentration of Cu in complete feedstuff with a moisture content of 12% is 15 mg/kg for ovines.

Ovine white liver disease (OWLD) is a disorder of the lipometabolism of sheep, usually growing lambs, and was first recorded in New Zealand in 1967 [21]. The trace element cobalt (Co) is an essential component for the synthesis of vitamin B_12_ (Cobalamin) by rumen microbes [22]. Thus, Co deficiency secondary results in a vitamin B_12_ deficiency. Sheep appear to be susceptible to Co deficiency and develop normocytic and normochromic anaemia, anorexia, reduced weight gains and photosensitivity [23,24]. Moreover, lacrimation, scaly ears, discolouration of the wool, cardiovascular lesions and cerebrocortical necrosis [25,26,27,28] have been associated with low dietary levels of Co. For adult sheep grazing Co deficient pastures the amount of Co necessary to ensure optimal growth is 0.08 mg/animal/d. The requirement for young, rapidly growing lambs is greater during the first few months and should be at least 0.2 mg/animal/day [29]. Clinicopathological abnormalities associated with Co deficiency are related to the inefficient metabolism of propionate [30]. Under normal conditions propionate is produced in the rumen, being metabolized to succinate via methylmalonate by the vitamin B_12_-dependent enzyme methylmalonyl-CoA mutase. Succinate enters the citric acid circle to provide glucose as an energy source [31]. A lack of Co and hence of vitamin B_12_ leads to insufficient β-oxidation of fatty acids which accumulate in the hepatocytes and result in hepatic lipidosis. Gross pathological findings of OWLD include a remarkably pale, beige-coloured and friable liver. Liver histopathology reveals fatty degenerations of hepatocytes and accumulations of ceroid lipofuscin in Kupffer cells [32]. However, it must be admitted that even in cases of sufficient Co supplementation by the diet a ruminal function disorder might results in a vitamin B_12_ deficiency due to the disfunction of the ruminal microbiota.

Deficiencies of vitamin E/Se are associated with nutritional myopathy with degenerative lesions of the skeletal muscles and heart muscle as well as hepatic necrosis, also known as white muscle disease [33]. Clinical manifestations associated with Vitamin E/Se deficiency in sheep include: ill-thrift in lambs, white muscle disease, lowered fertility and embryonic death, prematurity and perinatal death and abortions and immunosuppression [34,35].

This paper describes a combined occurrence of molybdenosis, secondary Cu deficiency, OWLD (Co deficiency) and selenium (Se) deficiency in a group of pastured ram lambs in Northern Germany. To the authors‘ knowledge, this is the first report of multiple trace element deficiencies in sheep in Germany for decades.

## 2. Materials and Methods

### 2.1. Case Presentation

This field investigation did not require official or institutional ethical approval because all samples were taken during routine diagnostic procedures to improve the herd health and were conducted in accordance with German animal welfare legislation and the EU Directive 2010/63/EU for animal experiments. All animals were handled according to high ethical standards and national legislation.

In October 2015 a German pedigree breeder with a small herd (8 ewes, 2 rams, 3 yearling ewes, and 17 lambs) of White Horned Heath sheep contacted the Clinic for Swine and Small Ruminants of the University of Veterinary Medicine Hannover, Foundation for a veterinary investigation into issues with lamb morbidity and mortalities. The farm is located in the North-Western part of Germany close to the Free City of Hamburg. The herd is managed outside at pasture year-round with access to an open stable at lambing time (March/April). Sheep are rotated regularly across two pastures during spring and summer. Pasture 1, based on sandy soil, is 2 hectares (ha) of a monoculture grass species (bluegrass, *Poa pratensis*) and sheep have been reared on this pasture for 3 years. Pasture 2 is a marshy soil type. It comprises 0.7 ha of diverse vegetation (pasture grass (*Poa pratensis*), timothy grass (*Phleum pretense*), meadow fescue (*Festuca pratensis*), velvet grass (*Holcus lanatus*), clover and legumes) and has been in use by sheep for 19 years. However, pasture grass dominates pasture 2 as well. Pasture 3 (0.5 ha) is used for lambing in March and April with the grazing of horses during the summer. No fertilizer is applied to pastures 1, 2 and 3. During the winter months (November until March) the herd is reared on temporary grassland which is fertilized (cattle manure and artificial fertilizer) and mowed regularly. Whilst sheep are at pasture, they are supplemented with a free-standing proprietary mineral block containing zinc (6400 mg/kg), manganese (1300 mg/kg), Co (20 mg/kg), iodine (100 mg/kg) and Se (24 mg/kg). The mineral block does not contain any Cu, Mo, S or Fe. Average intake, estimated by the owner based on block consumption, is approximately 2–4 g/animal/day, although significant intake deviations by some individuals, or in some periods cannot be ruled out. The recommendation for daily intake of the manufacturer is 20–30 g/animal/day. During the lambing period, the sheep are supplemented with ad libitum silage grass. The herd receives no concentrated feed and water access is freely available on all pastures (quality tested well water). Prior to these veterinary investigations, there was minimal veterinary practice contact. There was no history of preventative disease vaccination. The last deworming of the entire herd took place in spring 2015 with an oral moxidectin compound (0.2 mg/kg body weight, Cydectin^®^ 0.1%, Zoetis, Switzerland). No parasitological analysis or assessment of the FAMACHA© score was performed before the anthelmintic treatment by the local veterinarian. The average number of raised lambs/ewe/year was 1.3–1.5. Abortions and premature births had not been observed during the previous lambing periods. The shepherd reported that the lambs, especially the ram lambs aged 4–6 months, had shown poor growth performance over the previous two years (13 lambs were affected, 3 died). In 2014, the shepherd reported signs of emaciation and shaggy, pale wool. Post mortem investigation of one of these animals revealed cachexia, and the liver was pale and of friable consistency. Hepatic lipidosis was confirmed by histopathology. Additionally, in 2015 the farmer reported that the lambs had shown signs of photodermatitis (crusty and ulcerated skin over the labia, nasal planum, ears and distal parts of the limbs) during the summer months. Three of the 13 (23%) affected lambs in 2014 and 2015 died despite intensive treatment with oral moxidectin (0.2 mg/kg body weight, Cyedctin^®^ 0.1%, Zoetis, Switzerland) and oral triclabendazole (10 mg/kg body weight, Fasinex^®^ 5%, Novartis Tiergesundheit AG, Switzerland), supplementation of vitamin E/Se, and supplementation of B vitamins by the local veterinarian. Again no parasitological analysis or assessment of the FAMACHA© score or Se or Co status was performed by the local veterinarian prior to treatment. As the condition of the lambs did not improve despite the implemented treatment, the local veterinarian submitted one affected ram lamb to the clinic for further investigation on the 5th of October 2015. In autumn 2015, veterinary investigations performed by the Clinic of Swine and Small Ruminants included clinical examination of the entire herd (30 animals) was conducted.

### 2.2. Clinical Examination

An intensive clinical examination of a six-month-old entire ram lamb (15.5 kg) submitted to the clinic was performed on 5 October 2015.

### 2.3. Farm Visit

On the 29th of October 2015, a farm visit was made by an experienced sheep veterinarian (CH). The animals were reared in four different groups on four different pastures: group 1: Eight ram lambs kept on pasture 3 until slaughter; group 2: Three yearling ewes and one ram kept on temporary pasture grass until lambing; group 3: Eight ewes and one ram kept on temporary pasture grass until lambing; group 4: Nine female lambs kept on temporary pasture grass until slaughter. The whole herd was clinically examined. Blood samples, faecal samples and liver samples were taken to the clinic for further analysis. 

### 2.4. Further Diagnostics

An overview of samples gained for further diagnostics can be found in Table 1.

#### 2.4.1. Blood and Liver Samples

Blood samples were taken for further diagnostics (EDTA blood for haematology, Lithium-Heparin-Plasma and serum for biochemistry, serum for serology, Lithium-Heparin-Plasma and serum for investigation of mineral balance). In total 5 animals were sampled: 1. The ram lamb submitted to the clinic, 2. two 6-month-old entire ram lambs (one affected lamb showing clinical signs of emaciation and photodermatitis in healing and one lamb in apparently healthy condition), 3. two ewes (one emaciated seven-year-old ewe with shaggy wool and a three-year-old ewe in good condition). EDTA anticoagulated blood was used to analyse, haemoglobin concentration and white and red blood cell count (Haematology analyser, Celltag alpha, Nihon Kohden, Kleinmachnow, Germany). Packed cell volume was analysed by centrifugation. Erythrocyte indices (MCH, MCV, MCHC) were calculated. Heparin plasma was analysed for total protein, albumin, bilirubin, plasma enzyme activities of creatine kinase (CK), aspartate-amino-transferase (ASAT), glutamate dehydrogenase (GLDH), and gamma-glutamyl-transferase (GGT) were determined by routine laboratory biochemistry according to Bickhardt and König [36]. Serum samples were analysed for Selenium (Se), copper (Cu), and cobalt (Co) concentrations. The results of the liver fluke Elisa were evaluated according to the manufacturer´s instructions (IDEXX Fasciolosis Verification Test, IDEXX Laboratories, Ludwigsburg, Germany). Investigations of Se, Cu, Co and Mo concentrations in liver samples gained from 2 ewes of unknown age and health status slaughtered in 2014 and the ram lamb submitted to the clinic and euthanised due to grounds of animal welfare as well as plasma samples were performed by GFAAS and flame AAS (SOLAAR M, Thermo FisherScientific, Karlsruhe, Germany), respectively. Liver samples were digested using 4 mL HNO_3_ (65%) and 1ml H_2_O_2_ (30%) in a microwave digestion unit (Start, MLS GmbH, Leutkirch, Germany) before analysis [37,38].

#### 2.4.2. Faecal Samples

Pooled faecal samples were taken from each of the 4 different groups for endoparasitic investigation during the farm visit. Furthermore, a single faecal sample of the ram lamb submitted to the clinic was gained. For examining the faecal samples, a modified version of the combined sedimentation-flotation process according to Benedek [39] was used.

#### 2.4.3. Post Mortem Investigation

After three days of hospitalization and repeated clinical examination, the animal submitted to the clinic remained inappetent and was euthanized on the grounds of animal welfare. Post mortem investigation was performed at the Lower Saxony State Office for Consumer Protection and Food Safety, Food- and Veterinary Institute Braunschweig/Hannover. Gross necropsy examination was performed and tissue samples were collected from the major parenchymatous organs (lungs, heart, brain, liver, spleen, and kidneys), musculature and joints. Samples were fixed in 4% neutral buffered formalin and embedded in paraffin wax. Tissue sections (4 µm) were stained with haematoxylin and eosin (HE).

#### 2.4.4. Nutritional Assessment

Feed samples of the permanent pasture grass from pastures 1 and 2, the pastures on which the sheep were reared during the summer months in which the clinical signs were observed, were collected (one rubbish bag (capacity 60 L) per pasture; samples were obtained from all 4 corners and in a meandering pattern from the rest of the pasture) and submitted for feed analysis to the Institute of Animal Nutrition, University of Veterinary Medicine Hannover, Foundation. Dry Matter, Calcium (Ca), Phosphorus (P) and S contents were analysed according to the VDLUFA (book of methods) [40]. The reference ranges are set according to Kamphues et al. [41]. Sulphur was detected by using the Vario Max CNS (Elementar^®^). The ground samples were mixed with wolfram (VI) oxide, weighed in ceramic crucibles and incinerated in the Vario Max CNS by 1140 °C. The calcium content and trace elements were determined by atomic absorption spectrometry, phosphorus colourimetrically [42,43,44,45]. Molybdenum analysis was performed according to DIN EN ISO 17294-2 (2005) in an external laboratory (Sys Analytics Germany GmbH, Weilheim, Germany). Cobalt analysis was performed as described by Lange et al. [46]. In addition, water samples from both wells of these pastures were collected (2 L/well collected in clean plastic bottles) and also sent for analytical examination to the Institute of Animal Nutrition, University of Veterinary Medicine Hannover, Foundation.

## 3. Results

### 3.1. Clinical Examination

Clinical examination of a six-month-old entire ram lamb (15.5 kg) submitted to the clinic revealed poor body condition with a body condition score (BCS) of 1.0 out of 5 [47], and a shaggy, discoloured coat. A bilateral mucopurulent nasal and ocular discharge, as well as hyperaemia of the sub-conjunctival mucous membranes, was observed. Cardiopulmonary auscultation and palpation of superficial lymph nodes were unremarkable. Rectal body temperature was 38.0 °C and the breech dirtiness (DAG score [48]) was 0 out of 5. The FAMACHA© score was 3–4 on a scale ranging from 1–5 (pale mucous membranes). One rumen contraction in two minutes was noticed. Assessment of the claws, joints and external reproductive organs was unremarkable. Severe multifocal pinhead-sized, encrusted, hyperaemic skin lesions were observed around the labia, the nose, the eyes and the ears. Externally, the nasal planum, periorbital skin and peri-auricular skin was bilaterally swollen, warm and appeared painful to palpation. Crusting of the skin could not be removed without loss of substance. Pruritus was not present. Based on clinical examination, the key clinical signs of note were emaciation, conjunctivitis and photodermatitis.

### 3.2. Farm Visit

Clinical examination of the entire herd during the farm visit which took place on the 29th of October 2015 revealed that ram lambs (group 1) were generally in poor body condition (average BCS was 1.5). Two ram lambs showed signs of conjunctivitis and photodermatitis in healing. The rest of the herd appeared to be clinically in good health except for one older ewe (7 years old), which was in a poor body condition (BCS 2) and also showed shaggy, pale wool with an open fleece.

### 3.3. Further Diagnostics

#### 3.3.1. Blood and Liver Samples

The ram lamb admitted to the clinic (sample ID 1) showed a remarkable increase of liver enzyme activities (aspartate aminotransferase (ASAT), glutamate dehydrogenase (GLDH), gamma-glutamyltransferase (GGT)) as well as an increase of total bilirubin. Microcytic, hypochromic anaemia, slight hyperproteinaemia and hypoalbuminaemia were identified. Analysis of liver tissue revealed that lamb 1 showed a severe copper and cobalt deficiency. Se in serum was within the reference range. Two lambs were sampled during the farm visit (sample ID 2 and 3), both showed low copper concentrations in serum and slightly elevated GGT activity. Signs of anaemia were noted in the haematology profile of lamb 2 (sample ID 2). Serum samples from ewes 1 and 2 (sample IDs 4 and 5) were only investigated for Se and Cu. Ewe 1 showed a moderate Se deficiency. Analysis of livers from two slaughtered adult sheep (sample IDs 7 and 8) revealed both a Se and Co deficiency. A Cu deficiency was noted in sample ID 8 and high concentrations of Mo were found in both liver samples. A summary of all results gained from blood and liver samples can be found in Table 2.

#### 3.3.2. Faecal Samples

The single faecal sample of lamb 1, the pooled faecal sample of the group of ram lambs and the pooled faecal sample of the group of ewes and rams showed only low egg counts. The pooled faecal samples of the female lambs and yearling ewes revealed moderate egg counts for gastrointestinal nematodes (202 eggs/g faeces and 117 eggs/g faeces, respectively; Table 3). No gross signs of fasciolosis were evident following inspection of livers received from slaughtered lambs in the previous years. The liver fluke serology was performed on the serum sample of lamb 1 with a negative result. An overview of the results is displayed in Table 3.

#### 3.3.3. Post Mortem Investigation

Gross necropsy examination of lamb 1 (sample ID 1) revealed crusty and ulcerated skin over the nasal planum (region nasalis), ears and distal parts of the skin of all four limbs. The liver was friable with yellow discolouration. The remaining gross organ examination was unremarkable. Microscopical examination of the skin revealed a severe pustular to ulcerative dermatitis with orthokeratotic hyperkeratosis, multifocal epidermal hyperplasia and a mild multifocal histiocytic to suppurative infiltration. Hepatocytes showed severe accumulation of micro- and macrovesicular fat droplets with single-cell necrosis (fatty degeneration). Mild biliary hyperplasia with cholestasis and a mild periportal lymphocytic and histiocytic infiltration were also noted (Figure 1).

#### 3.3.4. Nutritional Assessment

Results of the nutritional assessments are listed in Table 4. Samples of grass from pastures 1 and 2, identified elevated S, Fe and Mo, and low Se and Ca concentrations. The Cu: Mo-ratio was <2:1 on both pastures. The analysis of the water samples was unremarkable (clear liquid, odourless, iron values <0.066 mg/L, the reference value [41]: <3 mg/L).

## 4. Discussion

Veterinary diagnostic investigations indicate a concomitant molybdenosis, severe secondary Cu deficiency, OWLD (Co deficiency) and Se deficiency in this pedigree sheep herd. Small numbers of investigated animals and samples limit the scope of interferences between the different minerals in this case report so that the conclusions concerning their interference should be taken with some caution.

The ram lamb which was admitted to the clinic showed emaciation, ill-thrift, discolouration of the wool, anaemia, lacrimation, conjunctivitis and photodermatitis. Liver enzyme activities (ASAT, GLDH, GGT) were distinctly elevated due to serious liver damage. Furthermore, total bilirubin was increased. Examination of liver samples revealed a severe Cu and Co deficiency. Se measured in a serum sample was just within the reference range, which might be explained by the fact that the group of ram lambs was treated with a vitamin E/Se compound subcutaneously one week prior to admission to the clinic, and 3 out of 5 animals tested for Se showed a Se deficiency. Hence, a Se deficiency before treatment might be assumed, In addition, this group of lambs was treated with B vitamins, oral moxidectin (0.2 mg/kg body weight, Cydectin^®^ 0.1%, Zoetis, Switzerland) and oral triclabendazole (10 mg/kg body weight, Fasinex^®^ 5%, Novartis Tiergesundheit AG, Switzerland) by the local veterinarian. Parasitological results of the faecal samples of this lambs were inconspicuous which is not surprising due to the performed deworming. A liver fluke serology was negative and post-mortem investigations showed no gross findings of fasciolosis. Cachexia, hepatic lipidosis and acute signs of photodermatitis in the area around the eyes, the ears and the nasal bridge were the main findings during necropsy.

As displayed in Table 2, 3 animals out of 5 showed Se deficiency and 1 animal was just within the reference range, 4 animals out of 7 showed Cu deficiency, 3 animals out of 3 showed Co deficiency and 2 animals out of 2 showed elevated Mo levels in a herd of 30 sheep. Due to the elevated Mo contents in two examined liver samples and the feed samples, a clinical picture of molybdenosis was assumed, and we conclude with caution that this led to secondary Cu deficiency. The farm is located 15 km away from a large Mo and Cadmium producing factory. Moreover, a coal fired power station is located nearby (distance: 30 km). It can be assumed, but not proven, that an intake of heavy metals due to the surrounding industry is possible. Typical clinical signs of secondary Cu deficiency were observed in this sheep herd, especially in the growing ram lambs. This was supported by blood and liver results that found evidence of a mild to severe Cu deficiency. MacPherson et al. [53] stated that the relation between Cu in the liver and blood was poor. These findings were proven by West et al. [54] which suggested that measuring concentrations of Cu in serum or plasma are not appropriate indicators of marginal Cu status in sheep as the decrease in concentrations of Cu in the blood occur later in the development of Cu deficiency in sheep than in cattle, so that many sheep with depleted stores of Cu in the liver show adequate concentrations of Cu in the serum. Hence, Cu concentrations measured in serum samples of sheep are often not very reliable. However, if the Cu concentrations measured in serum samples are low, a severe copper deficiency and depleted stores of Cu in the liver can be assumed. Generally, it is more useful to investigate liver samples if disorders in the Cu balance should be addressed to be aware of potential disorders in Cu supplementation at an early convenience. Liver samples can be gained of slaughtered or dead animals as well as by liver biopsy [37]. Cu deficiency is aggravated by high dietary amounts of S and Fe, which were also found in the examined pasture grass samples. The pastures (pasture 1 and 2), on which the herd grazes in spring and summer are not fertilized at all. Lambs are reared on pasture 1 and 2 dominated by monoculture pasture grass most of the time from birth until slaughter. By contrast, adult ewes and replacement yearling ewes are reared on fertilized and mowed pastures during the winter months. As the winter pastures are regularly fertilized by artificial fertilizer and cattle manure, which among others contents high levels of Cu [55], it can be assumed that the Cu deficiency, which arises during the summer months on the non-fertilized pastures, can be compensated by the adult sheep during the winter months. This might explain why clinical signs of Cu deficiency only appear in lambs aged 4–6 months, which at least for Cu, is unusual, as the pregnant ewes can fill up their mineral balance during pregnancy so that the intrauterine development of the lambs is not disturbed. First clinical signs only appear after weaning when the lambs are fed by non-fertilized forage of pasture 1 and 2. Thus, the peak of clinical signs only becomes obvious in the 4–6 months aged lambs. Management differences according to production stage and age of animals on this herd, and the different mineral nutrient requirements of growing ruminant animals might explain the owner’s reports of poor performance in growing lambs. In general, adult sheep did not show overt clinical signs of deficiency during the herd visit. Although, deficiencies were identified in samples collected from adult slaughtered sheep, to date no reports of newborn lambs with signs of Cu deficiency characterized by hindlimb ataxia (colloquially known as “swayback syndrome”) [56] have been reported in this herd.

A severe Co deficiency was found in three animals. The lamb admitted to the clinic showed typical clinical signs of OWLD including anaemia, anorexia, reduced weight gain, photosensitivity, lacrimation, scaly ears and discolouration of the wool [23,24]. Liver histopathological examinations revealed hepatic lipidosis, which is also typical for OWLD [32] due to the disorder of lipometabolism caused by Co and secondary Vitamin B_12_ deficiency. Due to the severe liver damage associated with OWLD, it is not uncommon to identify elevated liver enzymes (GGT, GLDH, AST), signs of emaciation and hepatic photodermatitis in affected animals [57]. Ulvund [58] found that in Norway especially the coastal areas are affected by Co deficiency. These finding could be proven by Sivertsen and Plassen [59]. Hence, it might be assumed that light and sandy soils are predisposed to low Co amounts. The absorption of Co through the plants depends on the pH value of the soil, and this decreases with increasing pH values. Additionally, the Co amount of the plants depends on the plant species. Grasses show the lowest Co content, whereas legumes, clover and lucerne contain 3–5 times more Co compared to grasses [60]. Pastures 1 and 2 which are used for growing lambs during summer are dominated by grass as a monoculture. Low Co contents in these forages are therefore not surprising.

Se deficiency is widespread in German sheep herds. Humann-Ziehank et al. [38] found that more than one-third of the investigated herds showed Se deficiency. Hence, if no mineral feed is provided or the Se content of the mineral feed is low then there is a risk for Se deficiency in grazing sheep herds. In this herd, the farmer reported that the sheep take only approximately 2–4 g mineral feed per animal per day of the offered licking mineral block. To assume an adequate mineral supply, adult sheep of approximately 60 kg should be fed 20–30 g/animal/day according to the manufacturer’s instructions. The licking block is presented in a 2 kg bucket with ad libitum access on the ground with no protection against bad weather conditions. Hence, much of the licking stone is washed out by rain. Moreover, contaminations due to urine, faeces and dirt might lead to diminished intake of mineral feed. In this herd, 3 tested animals showed a Se deficiency and one animal was just within the reference range. The group of ram lambs had been treated with a vitamin E/Se compound subcutaneously one week prior to admission of lamb 1 to the clinic. Therefore, given the long-term nature of Se, it is unknown but was assumed that Se deficiency arose pre-treatment. Se deficiency might lead to nutritional muscular dystrophy (white muscle disease) which might clinically result in ill thrift in lambs, which could be observed in the ram lambs of this herd.

As this herd is reared on pastures close to Hamburg, which is a large industrial city, an input of heavy metals from surrounding industries and coal-fired power stations in combination with a non-fertilized, mostly sandy soil and monoculture pasture grass might be a possible explanation for the disorders in the mineral balance of this herd. Hence, an input of heavy metals from surrounding industries and coal mining and a prior lack of veterinary input on the herd was considered to contribute to the severe and concomitant trace element deficiencies identified.

As a result of veterinary diagnostic investigations, injecting 3 mL vitamin E/Se per animal [Vitamin E-Selen (100 mg/mL + 0.658 mg/mL), CP-Pharma, Burgdorf, Germany] and vitamin B compounds (3 mL/animal, VITAMIN-B-KOMPLEX pro inj., Serumwerk Bernburg, Bernburg, Germany) subcutaneously and providing a mineral feed containing Co, Cu and higher amounts of Se fed at a dose of at least 20–30 g/animal/day according to the manufacturer’s instructions the ram lambs recovered quickly and achieved their slaughter weight (35 kg live weight on average), even though this was achieved 4–6 weeks later than expected. Raisbeck et al. [12] showed that moderate Cu supplementation permitted cows to graze on pastures heavily contaminated with Mo with no adverse effects on general health or reproduction. Hence, a mineral feed for sheep containing Cu should be provided from now on being freely accessible for all animals at any time. To ensure this, the farmer was advised to build roofed facilities at head value and feed 20–30 g/animal/day of a commercial mineral feed in powdered or pelletized form. To monitor the Cu intake of the herd the farmer was advised to send in liver samples of slaughtered animals every 3–6 months so that the risk of an accidental Cu intoxication can be minimized. In addition, all newborn lambs will from now on be treated with vitamin E/Se and vitamin B compounds within the first days of life. Se and Co statuses will also be addressed from liver samples of slaughtered animals regularly to monitor the general mineral status of the herd.

## 5. Conclusions

This case report shows that several mineral deficiencies might be present at the same time in a sheep herd and that even under field conditions a detailed investigation is possible. Cases of poor growth performance in lambs should be investigated taking several mineral deficiencies, particularly Co, Cu and Se, into account. Clinical examination can give often only suspected diagnoses. To access possible mineral deficiencies, a nutritional assessment should be performed. Sampling should not only include blood, but also liver samples (biopsies, slaughter samples, or post mortem samples).

## Figures and Tables

**Figure 1 animals-11-01864-f001:**
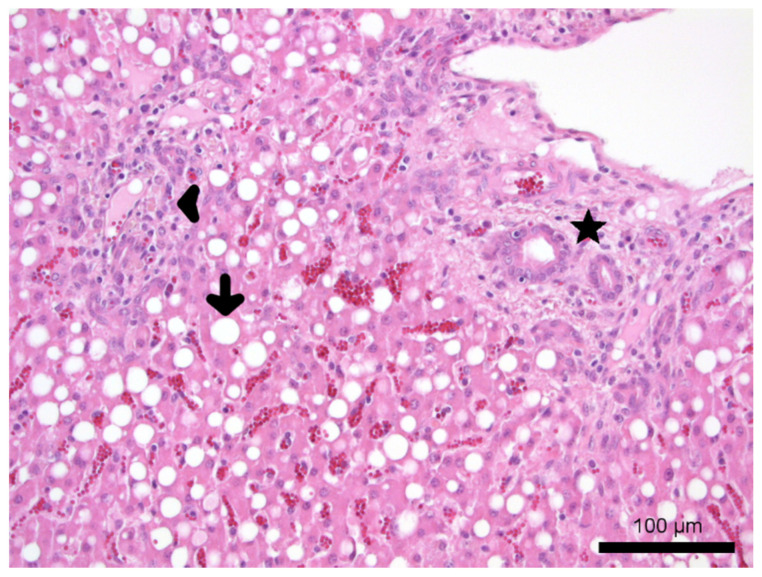
Histopathology of the liver: liver, HE, 20× magnification, hepatic lipidosis composed of fat droplets (arrow) within hepatocytes, a mild periportal lymphocytic and histiocytic infiltration (arrowhead) and mild biliary hyperplasia (asterisk). The sample was gained from the ram lamb submitted to the clinic for further investigations and euthanized on grounds of animal welfare.

**Table 1 animals-11-01864-t001:** Animals sampled for further investigations.

Id	Sample Id	Age	Sex	Health Status	Tissue Examined	Date
lamb 1– presented in the clinic	1	6 months	m	emaciation, photodermatitis, conjunctivitis, apathy	blood samples, faecal sample, post mortem investigation, liver samples	5/10/15
lamb 2– farm visit	2	6 months	m	unremarkable	blood samples, pooled faecal sample	29/10/15
lamb 3– farm visit	3	6 months	m	emaciation, photodermatitis in healing	blood samples, pooled faecal sample	29/10/15
ewe 1– farm visit	4	7 years	f	emaciation, shaggy, pale wool	blood samples, pooled faecal sample	29/10/15
ewe 2– farm visit	5	3 years	f	unremarkable	blood samples, pooled faecal sample	29/10/15
liver 1– farm visit, frozen	6	adult, concrete age u	f	u	liver samples	slaughtered in autumn 2014
liver 2– farm visit, frozen	7	adult, concrete age u	f	u	liver samples	slaughtered in autumn 2014

u = unknown, m = male, f = female.

**Table 2 animals-11-01864-t002:** Blood and liver results of tested sheep.

Sample ID	1	2	3	4	5	6	7
Selenium *** (s: 80–500 µg/L l: 0.25–1.5 mg/kg FM)	s: 83.5	s: 125.2	s: 112.3	**s: 60.9 ↓**	s: 125.3	**l: 0.109 ↓**	**l: 0.099 ↓**
Copper *** (s: 7–24 µmol/L; l: 10–120 mg/kg FM)	s: 10.9 **l: 1.45 ↓**	**s: 5.8 ↓**	**s: 5.7 ↓**	s: 13.7	s: 12.3	l: 24.2	**l: 9.1 ↓**
Cobalt *** (l: 0.025–0.085 mg/kg FM)	**l: 0.008 ↓**	-	-	-	-	**l: 0.015 ↓**	**l: 0.020 ↓**
Molybdenum *** (l: 1.5–6 mg/kg DM)	-	-	-	-	-	**l: 7 ↑**	**l: 8.5 ↑**
Haemoglobin * (90–140 g/L)	**84 ↓**	90	109	-	-	-	-
Packed Cell volume * (0.27–0.41 L/L)	**0.24 ↓**	0.3	0.36				
MCH * (13–14 pg)	**10.4 ↓**	**10.5 ↓**	**10.7 ↓**	-	-	-	-
MCV * (34–46 fL)	**32 ↓**	35.1	35.3				
MCHC * (290–340 g/L)	323	300	303				
Bilirubin ** (0–10 µmol/L)	**13.13 ↑**	-	-	-	-	-	-
Protein ** (52–70 g/L)	**71.9 ↑**	62.5	66.2				
Albumin ** (27–39 g/L)	**24.2 ↓**	36	31.8	-	-	-	-
ASAT ** (30–80 U/L)	**589 ↑**	63	47	-	-	-	-
CK ** (10–230 U/L)	87	78	100	-	-	-	-
GLDH ** (2–12 U/L)	**525 ↑**	7	9	-	-	-	-
GGT ** (5–32 U/L)	**135 ↑**	**34 ↑**	**35 ↑**	-	-	-	-

Sample ID according to Table 1; s: serum; l: liver, MCH: Mean Corpuscular Haemoglobin, MCV: Mean Corpuscular Volume, MCHC: Mean Corpuscular Haemoglobin Concentration. The number in brackets marks the reference range according to Weiss and Wardrop [49] *, Bickardt et al. [50] **, and Puls (1995) [51] ***. Values, which are not reported (hyphen), were not tested. The deviating values are indicated in bold and marked with ↓ for values that are too low and ↑ for elevated values.

**Table 3 animals-11-01864-t003:** Results of faecal samples of the four different groups (pooled, farm visit) and lamb 1 (individual, clinic).

Faecal Sample	Coccidial Oocysts/g Faeces	Gastrointestinal Nematodes Eggs/g Faeces	Nematodirus Eggs/g Faeces	Trichuris	Capillaria	Strongyloides Eggs/g Faeces	Moniezia spp.	*Dicrocoelium dentriticum*	*Fasciola hepatica*
lamb 1	5	45	0	−	−	3	−	−	−
ram lambs	17	7	33	+	−	10	−	−	−
female lambs	12	202	0	+	−	17	−	−	−
yearling ewes	5	117	2	−	−	4	−	−	−
ewes and rams	0	4	0	−	−	3	−	−	−

+ = positive; − = negative.

**Table 4 animals-11-01864-t004:** Results of the feed analysis of pastures 1 and 2.

	Reference Values	Pasture 1	Pasture 2
DM (g/kg original substance)	-	205	214
Ca (g/kg DM)	7.1 *	**4.66 ↓**	**5.46 ↓**
P (g/kg DM)	3.4 *	4.02	3.55
S (g/kg DM)	2.58 **	**3.79 ↑**	2.60
Fe (mg/kg DM)	50–280 ***	**404 ↑**	**531 ↑**
Cu (mg/kg DM)	5–10 ***	9.26	8.61
Co (mg/kg DM)	0.1–0.2 *	0.124	0.16
Se (mg/kg DM)	0.4–1 ***	**0.112 ↓**	**0.018 ↓**
Mo (mg/kg DM)	0.5–3.5 ***	**11.3 ↑**	**6.34 ↑**
Cu: Mo-ratio	Set point value: 6:1, 2–3:1 limits of acceptability, <2:1 toxic ****	**0.8:1 ↓**	**1.4:1 ↓**

DM: dry matter, Ca: Calcium, P: Phosphorus, S: Sulphur, Fe: Iron, Cu: Copper, Co: Cobalt, Se: Selenium, Mo: Molybdenum; The reference ranges are set according to Kamphues et al. [41] *, Dohm [52] **, Puls [51] ***, and Laiblin and Stöber [7] ****. The deviating values are indicated in bold and marked with ↓ for values that are too low and ↑ for elevated values.

## Data Availability

Data is contained within the article.

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
