# Peer review of "A Case of Concurrent Molybdenosis, Secondary Copper, Cobalt and Selenium Deficiency in a Small Sheep Herd in Northern Germany"

_animals, 2021, doi:10.3390/ani11071864_

Round 1
Reviewer 1 Report
"...this is the first report of multiple trace element deficiencies in sheep Germany." Are you sure? Seems unlikely. Maybe this is the first report from the specific village or precise location, breed and production system? But multiple deficiencies are neither new or unknown in the rest of the world; why would the be unknown in Germany? Please check again.
The term "gimmer". Most readers would better recognize the term "yearling ewe" instead.
What are the copper and molybdenum contents of the mineral block? How much of that mineral block was consumed? Do you think the generous Zn content of the block affected Cu or Co absorption?
Were parasites found and identified, or something similar to FAMACHA scoring of eye and membrane color before attempting worming with moxidectine and and other compounds?
"Abortions or premature births were not observed during the previous lambing periods." No abortions at all over about 19 years of sheep on these pastures?
No responses to vitamin E/Se or chemical wormers? Were there indications of internal parasites to begin with? I don't thing you can assume pre-treatment, pre-study Se deficiencies, since you have no samples or chemical data from that period of time.
"Rumen turnover time" This term is used to express the passage relative to retention of rumen contents. You might want to avoid confusion by writing: "We observed one rumen contraction per two minutes."
Please detail the specific methods used to measure plasma Se, Co, Cu (and Mo, if measured) with references. Also, please be more specific than "routine laboratory biochemistry" in describing the methods for liver enzymes with references.
For feed and water analyses, please specify and characterize the methods used to measure Ca, P, S, Co, Cu, Mo (and Zn?).
Table 2:
Selenium: Four of the sheep had adequate selenium based on blood values. one was low by blood values, two were low by liver values.
Copper: Serum copper values for Cu are known to be unreliable indicators of ovine copper status (as pointed out by the authors themselves in the discussion).That said, 3/5 of the serum samples were in the "normal" range, one liver sample was really low, but the the other two liver samples were borderline or adequate. I am willing to believe that some of then animals in this flock were copper deficient based on the color and condition of their fleeces, the blood tissue numbers seem inconclusive for the group.
Cobalt: The three liver samples indicated serious Co deficiency, too bad there are no figures for more of the sheep.
Molybdenum: There are only two liver samples available, but they indicate some seriously elevated Mo in the system.
While the appearance of some the the animals and two or three livers would be consistent with suppression of Co uptake by Mo and maybe Cu uptake, as well, the very few observations available should make one cautious about making inferences beyond the individual animals and soil/plants/feeds observed.
Figure 1. Which sheep provided the liver section shown?
Table 4. How were feed Co, Cu, Mo, P, S, Fe, Se, Ca actually measured? Give enough detail so similar analyses could be applied to repeat this work.
Please put in a row for actual Co content of the feed so readers can compare it to other minerals themselves. It is OK to display Co:Mo ratios, too, if you want to keep those.
Do you have figures for the soil mineral content? That could help with figuring out sources of feed minerals.
Contributions of local industrial sources to high Mo make sense, but the research design presented does not address that directly. For this specific site, were excess levels of Cd, Pb and other heavy metals found in the soil air, water, plants, feed or sheep? How about Zn?
Why are on-farm post mortem examinations and tissue sample collections not allowed? It would seem that producers could be educated to dissect their animals, photograph the results and collect useful samples for submission. Are vets not allowed to do on-farm post mortem, either, or just farmers?
Author Response
Reply to Reviewer 1
Dear Reviewer 1,
Thank you very much for editing our paper. We tried to address your comments as best as we could. Please find our answers to your questions below.
"...this is the first report of multiple trace element deficiencies in sheep Germany." Are you sure? Seems unlikely. Maybe this is the first report from the specific village or precise location, breed and production system? But multiple deficiencies are neither new or unknown in the rest of the world; why would the be unknown in Germany? Please check again.
The authors could not find a report in which all of these trace element deficiencies were described at the same time in one sheep flock in Germany and the literature of described trace element deficiencies was rather old. However, we changed the above mentioned sentence to “To the authors` knowledge, this is the first proven report of multiple trace element deficiencies in sheep in Germany since decades.” (ll 31/32)
The term "gimmer". Most readers would better recognize the term "yearling ewe" instead.
The term gimmer was replaced by “yearling ewe” throughout the document as suggested.
What are the copper and molybdenum contents of the mineral block? How much of that mineral block was consumed? Do you think the generous Zn content of the block affected Cu or Co absorption?
The Cu and Mo contents of the mineral block was 0. This was complemented in the document (ll. 145-147). We do not think that the generous Zn content did effect the Cu or Co absorption in this particular flock as the average daily intake described by the animal owner was only 2-4 g/animal/ day which is far less that recommended by the manufacturer stating that 20-30g/animal/day should be applied.
Were parasites found and identified, or something similar to FAMACHA scoring of eye and membrane color before attempting worming with moxidectine and and other compounds?
No, unfortunately no parasitological analysis or assessment of the FAMACHA score were performed prior to treatment by the local veterinarian. This fact was complemented in the document where necessary.
"Abortions or premature births were not observed during the previous lambing periods." No abortions at all over about 19 years of sheep on these pastures?
No, the farmer never had any issues with abortions or pre-mature births in this flock according to his records.
No responses to vitamin E/Se or chemical wormers? Were there indications of internal parasites to begin with? I don't thing you can assume pre-treatment, pre-study Se deficiencies, since you have no samples or chemical data from that period of time.
No, the animal submitted to the clinic did not show any improvement of the general health status despite the above-mentioned treatment and therefor was euthanised due the grounds of animal welfare. Seeing the results of PM examination, the poor health status was not surprising as the liver damage was severe. You are right, that we do not have any data of the animal pre-treatment, however, as the Se status in the serum sample of the submitted ram lamb is still just within the reference range despite correct treatment with Vit E/selenium 1 week prior to admission to the clinic, a certain deficiency might be assumed. Moreover, 3 additional animals not treated with Vit E/selenium prior to Se analysis also showed a Se deficiency in investigated serum or liver samples. Hence, a general history of Se deficiency in the flock is present.
"Rumen turnover time" This term is used to express the passage relative to retention of rumen contents. You might want to avoid confusion by writing: "We observed one rumen contraction per two minutes."
The term rumen turnover time was changed as suggested by the reviewer (ll. 276-277).
Please detail the specific methods used to measure plasma Se, Co, Cu (and Mo, if measured) with references. Also, please be more specific than "routine laboratory biochemistry" in describing the methods for liver enzymes with references.
The methods for liver enzymes analysis we used are described in detail by Bickhardt und König (1999), the citation was added. The methods we used for analysis of Se, Co, and Cu concentrations in plasma are described in detail by Humann et al. 1999, and Humann-Ziehank et al. 2013. Originally the description for the element-analysis was written for serum, but in fact the analysis were made for heparin plasma. The word “serum” was changed to “plasma” in line 227. The citations are added.
For feed and water analyses, please specify and characterize the methods used to measure Ca, P, S, Co, Cu, Mo (and Zn?).
Methods used to measure Ca, P, S, Co, Cu and Mo for nutritional assessment of the pasture samples were added (ll. 254-264). Zn was not measured neither from feed or water samples, nor from blood or liver samples.
Table 2:
Selenium: Four of the sheep had adequate selenium based on blood values. one was low by blood values, two were low by liver values.
This is correct. 3 out of 5 animals tested for Se showed a Se deficiency. Moreover, the ram lamb submitted to the clinic was just within the reference range.
Copper: Serum copper values for Cu are known to be unreliable indicators of ovine copper status (as pointed out by the authors themselves in the discussion). That said, 3/5 of the serum samples were in the "normal" range, one liver sample was really low, but the the other two liver samples were borderline or adequate. I am willing to believe that some of then animals in this flock were copper deficient based on the color and condition of their fleeces, the blood tissue numbers seem inconclusive for the group.
It is correct, that the Cu status should be analysed in liver samples rather than in serum samples. However, if the Cu content in the serum samples is far below the reference value, one can assume a severe Cu deficiency as the serum Cu level decreases far later than the liver Cu level. Hence, in cases of very low Cu values in serum samples those are adequate to point out a Cu deficiency. However, values that are within the reference range or elevated are not reliable and should always be confirmed by analysis liver samples. The authors stated that 4 out of 7 animals investigated for Cu show a Cu deficiency in serum or liver samples. 3 of these 4 samples are severe and clinical as well as haematological results (microcytic, hypochromic anemia) support the diagnosis of Cu deficiency.
Cobalt: The three liver samples indicated serious Co deficiency, too bad there are no figures for more of the sheep.
The authors would have liked to examine more liver samples for Co. However, as there was no founding for this investigation and the flock examined was a small-scale pedigree sheep flock of which the owner was not supportive in taking more liver samples, we were not able to obtain any more liver samples form these sheep.
Molybdenum: There are only two liver samples available, but they indicate some seriously elevated Mo in the system.
This is correct. The reason for the small number of samples examined are the same as stated under Cobalt.
While the appearance of some the the animals and two or three livers would be consistent with suppression of Co uptake by Mo and maybe Cu uptake, as well, the very few observations available should make one cautious about making inferences beyond the individual animals and soil/plants/feeds observed.
It is right that the general number of samples analysed was low due to the reasons described above. However, the authors think that this severe and combined deficiency of trace elements in this particular pedigree sheep flock was quite unique and although it was rather a detailed case report than a planned research project, the data gained might be useful to other sheep veterinarians, especially those working in veterinary practice, to have an idea how to assess such serve and combined mineral deficiencies under field conditions.
Figure 1. Which sheep provided the liver section shown?
The histopathological examinations of Figure 1 belong to the ram lamb submitted to the clinic for further diagnostics which was euthanised to reasons of animal welfare (sample ID 1). A sentence to clarify to which animal this figure belongs was added below Figure 1.
Table 4. How were feed Co, Cu, Mo, P, S, Fe, Se, Ca actually measured? Give enough detail so similar analyses could be applied to repeat this work.
The methods used for nutritional assessments of the pasture grass samples were described under M&M as pointed out above.
Please put in a row for actual Co content of the feed so readers can compare it to other minerals themselves. It is OK to display Co:Mo ratios, too, if you want to keep those.
A row for the Co content was added. The row with the Cu:Mo ratio was kept (here was a mistake made by the authors as it should be Cu:Mo ratio and not Co:Mo ration; this mistake was corrected in Table 4) as this ratio is used to describe the occurrence of molybdenosis.
Do you have figures for the soil mineral content? That could help with figuring out sources of feed minerals.
No, soil was not investigated. Only pasture grass of pastures 1 and 2 as described in the manuscript due to cost restrictions.
Contributions of local industrial sources to high Mo make sense, but the research design presented does not address that directly. For this specific site, were excess levels of Cd, Pb and other heavy metals found in the soil air, water, plants, feed or sheep? How about Zn?
As described above, this paper is rather supposed to be a detailed case report than a planned research approach. Due to lack of funding no other heavy metals were investigated. However, this is of course a very good idea for further planned research projects addressing those issues in the surrounding of the industrial city of Hamburg.
Why are on-farm post mortem examinations and tissue sample collections not allowed? It would seem that producers could be educated to dissect their animals, photograph the results and collect useful samples for submission. Are vets not allowed to do on-farm post mortem, either, or just farmers?
In Germany is it forbidden by law to perform on farm PM investigations regardless whether they are performed by the farmer or a veterinarian. All carcasses for PM regardless which animal species need to be send in to specialised labs. Some bigger veterinary practices have their own necropsy room approved by the local veterinary authorities but this is not a standard. Thus, PM investigations are rather expensive and therefor not performed as often as they should.
Reviewer 2 Report
Dear authors,
The clinical case presented here shows a severe deficiency disorder in a pedigree sheep German flock. The clinical study is well directed and oriented, and the results are conclusive. The introduction provides sufficient background, the discussion is well addressed, and the conclusion is supported by the results. However, as different tests have been carried out on the analysed animals and many different results are shown, it is a bit confusing. The organisation of the tables is also somewhat unclear since Table 2, which corresponds to results, is shown in material and methods, and mention is made of Table 4 that appears at the end of the text. The order of the tables should be rearranged and mentioned wherever they are displayed. The results should have the same order and statement as the material and methods, explaining what was obtained in each of the sections. Perhaps it would be useful to create a summary table with the clinical and laboratory alterations collected from each animal.
In addition, it could be interested to add a picture of the facial lesions if available by the authors.
In line 324 of the discussion, the number of animals that showed Cu deficiency is missing.
In line 412: a word is missing: 4-6 weeks “later” than expected.
Line 440: word missing: taking samples “in” the right way.
The conclusions show an interesting and pedagogical point of view on how to approach a case of mineral deficiencies that I think is appropriate. However, section 2 should explain which serological tests it refers to.
Author Response
Reply to Reviewer 2
Dear Reviewer 2,
Thank you very much for editing our paper. We tried to address your comments as best as we could. Please find our answers to your questions below.
The organisation of the tables is also somewhat unclear since Table 2, which corresponds to results, is shown in material and methods, and mention is made of Table 4 that appears at the end of the text. The order of the tables should be rearranged and mentioned wherever they are displayed. The results should have the same order and statement as the material and methods, explaining what was obtained in each of the sections. Perhaps it would be useful to create a summary table with the clinical and laboratory alterations collected from each animal.
You are absolutely right that the organisation of the Tables and Figures was confusing. The organisation was re-arranged and the Tables and Figures are now displayed where mentioned in the text. Furthermore, the M&M and the Results sections do now have the same sub-headings for a better transparency.
In addition, it could be interested to add a picture of the facial lesions if available by the authors.
Unfortunately we have no pictures of these lesions.
In line 324 of the discussion, the number of animals that showed Cu deficiency is missing.
The missing number was added.
In line 412: a word is missing: 4-6 weeks “later” than expected.
The missing word was added as suggested.
Line 440: word missing: taking samples “in” the right way.
The missing word was added as suggest.
The conclusions show an interesting and pedagogical point of view on how to approach a case of mineral deficiencies that I think is appropriate. However, section 2 should explain which serological tests it refers to.
In section 2 the serological tests which are meant were now clearly stated by the authors.
Reviewer 3 Report
- This study was attempted to investigate the mineral deficiency in a small pedigree flock of White Horned Heath sheep in Germany. However, there is different mineral requirement for animals in different physiological stage. Therefore, the results is needed analyzed according to different physiological stage, and the statistical method should be described.
- The conclusion should be rewritten, bringing out more significant findings.
Author Response
Sorry, but I can not download the report of reviewer 3.
Kind regards,
Round 2
Reviewer 1 Report
|
Reply to Reviewer 1
“Concurrent molybdenosis, secondary copper deficiency, cobalt and
Dear Reviewer 1, Thank you very much for editing our paper. We tried to address your comments as best as we could. Please find our answers to your questions below. "...this is the first report of multiple trace element deficiencies in sheep Germany." Are you sure? Seems unlikely. Maybe this is the first report from the specific village or precise location, breed and production system? But multiple deficiencies are neither new or unknown in the rest of the world; why would the be unknown in Germany? Please check again. |
|
The authors could not find a report in which all of these trace element deficiencies were described at the same time in one sheep flock in Germany and the literature of described trace element deficiencies was rather old. However, we changed the above mentioned sentence to “To the authors` knowledge, this is the first proven report of multiple trace element deficiencies in sheep in Germany since decades.” (ll 31/32) OK The term "gimmer". Most readers would better recognize the term "yearling ewe" instead. The term gimmer was replaced by “yearling ewe” throughout the document as suggested OK What are the copper and molybdenum contents of the mineral block? How much of that mineral block was consumed? Do you think the generous Zn content of the block affected Cu or Co absorption? The Cu and Mo contents of the mineral block was 0. This was complemented in the document (ll. 145-147). We do not think that the generous Zn content did effect the Cu or Co absorption in this particular flock as the average daily intake described by the animal owner was only 2-4 g/animal/ day which is far less that recommended by the manufacturer stating that 20-30g/animal/day should be applied. OK Were parasites found and identified, or something similar to FAMACHA scoring of eye and |
|
membrane color before attempting worming with moxidectine and other compounds? No, unfortunately no parasitological analysis or assessment of the FAMACHA score were performed prior to treatment by the local veterinarian. This fact was complemented in the document where necessary. OK "Abortions or premature births were not observed during the previous lambing periods." No abortions at all over about 19 years of sheep on these pastures? No, the farmer never had any issues with abortions or pre-mature births in this flock according to his records. [That would be very impressive.] |
|
No responses to vitamin E/Se or chemical wormers? Were there indications of internal parasites to begin with? I don't thing you can assume pre-treatment, pre-study Se deficiencies, since you have no samples or chemical data from that period of time. No, the animal submitted to the clinic did not show any improvement of the general health status despite the above-mentioned treatment and therefor was euthanised due the grounds of animal welfare. Seeing the results of PM examination, the poor health status was not surprising as the liver damage was severe. You are right, that we do not have any data of the animal pre-treatment, however, as the Se status in the serum sample of the submitted ram lamb is still just within the reference range despite correct treatment with Vit E/selenium 1 week prior to admission to the clinic, a certain deficiency might be assumed. Moreover, 3 additional animals not treated with Vit |
|
E/selenium prior to Se analysis also showed a Se deficiency in investigated serum or liver samples. Hence, a general history of Se deficiency in the flock is present. OK. Sorry, I missed that part.
"Rumen turnover time" This term is used to express the passage relative to retention of rumen contents. You might want to avoid confusion by writing: "We observed one rumen contraction per two minutes." The term rumen turnover time was changed as suggested by the reviewer (ll. 276-277). OK
Please detail the specific methods used to measure plasma Se, Co, Cu (and Mo, if measured) with references. Also, please be more specific than "routine laboratory biochemistry" in describing the methods for liver enzymes with references. The methods for liver enzymes analysis we used are described in detail by Bickhardt und König (1999), the citation was added. The methods we used for analysis of Se, Co, and Cu concentrations in plasma are described in detail by Humann et al. 1999, and Humann-Ziehank et al. 2013. Originally the description for the element-analysis was written for serum, but in fact the analysis were made for heparin plasma. The word “serum” was changed to “plasma” in line 227. The citations are added. OK, thanks!
For feed and water analyses, please specify and characterize the methods used to measure Ca, P, S, Co, Cu, Mo (and Zn?). |
Methods used to measure Ca, P, S, Co, Cu and Mo for nutritional assessment of the pasture samples were added (ll. 254-264).
OK
Zn was not measured neither from feed or water samples, nor from blood or liver samples.
Table 2:
Selenium: Four of the sheep had adequate selenium based on blood values. one was low by blood values, two were low by liver values.
This is correct. 3 out of 5 animals tested for Se showed a Se deficiency. Moreover, the ram lamb submitted to the clinic was just within the reference range.
2
|
Copper: Serum copper values for Cu are known to be unreliable indicators of ovine copper status (as pointed out by the authors themselves in the discussion). That said, 3/5 of the serum samples were in the "normal" range, one liver sample was really low, but the the other two liver samples were borderline or adequate. I am willing to believe that some of then animals in this flock were copper deficient based on the color and condition of their fleeces, the blood tissue numbers seem inconclusive for the group. It is correct, that the Cu status should be analysed in liver samples rather than in serum samples. However, if the Cu content in the serum samples is far below the reference value, one can assume a severe Cu deficiency as the serum Cu level decreases far later than the liver Cu level. Hence, in cases of very low Cu values in serum samples those are adequate to point out a Cu deficiency. |
|
However, values that are within the reference range or elevated are not reliable and should always be confirmed by analysis liver samples. The authors stated that 4 out of 7 animals investigated for Cu show a Cu deficiency in serum or liver samples. 3 of these 4 samples are severe and clinical as well as haematological results (microcytic, hypochromic anemia) support the diagnosis of Cu deficiency. Good points. But given the very small number of animals, you still might want to express some caution.
Cobalt: The three liver samples indicated serious Co deficiency, too bad there are no figures for more of the sheep. The authors would have liked to examine more liver samples for Co. However, as there was no founding for this investigation and the flock examined was a small-scale pedigree sheep flock of which the owner was not supportive in taking more liver samples, we were not able to obtain any more liver samples form these sheep. More real-world limitations on sample size. But this missing data does call on us to be cautious with broad inferences.
Molybdenum: There are only two liver samples available, but they indicate some seriously elevated Mo in the system. This is correct. The reason for the small number of samples examined are the same as stated under Cobalt. While the appearance of some the animals and two or three livers would be consistent with suppression of Co uptake by Mo and maybe Cu uptake, as well, the very few observations available should make one cautious about making inferences beyond the individual animals and |
|
soil/plants/feeds observed. It is right that the general number of samples analysed was low due to the reasons described above. However, the authors think that this severe and combined deficiency of trace elements in this particular pedigree sheep flock was quite ? unique, [Unique is an absolute term.] [again, small numbers and missing data limit the scope of inferences. One might note that the few samples available are consistent with some of the deficiencies. And it is possible that more data from similar cases with more complete analyses would complete the picture. Your findings do indicate that vets and farmers should be on the lookout for situations like this.
and although it was rather a detailed case report than a planned research project, the data gained might be useful to other sheep veterinarians, especially those working in veterinary practice, to have an idea how to assess such serve and combined mineral deficiencies under field conditions. OK
Figure 1. Which sheep provided the liver section shown? |
3
|
The histopathological examinations of Figure 1 belong to the ram lamb submitted to the clinic for further diagnostics which was euthanised to reasons of animal welfare (sample ID 1). A sentence to clarify to which animal this figure belongs was added below Figure 1. OK
Table 4. How were feed Co, Cu, Mo, P, S, Fe, Se, Ca actually measured? Give enough detail so similar analyses could be applied to repeat this work. The methods used for nutritional assessments of the pasture grass samples were described under M&M as pointed out above. Please put in a row for actual Co content of the feed so readers can compare it to other minerals |
|
themselves. It is OK to display Co:Mo ratios, too, if you want to keep those. A row for the Co content was added. The row with the Cu:Mo ratio was kept (here was a mistake made by the authors as it should be Cu:Mo ratio and not Co:Mo ration; this mistake was corrected in Table 4) as this ratio is used to describe the occurrence of molybdenosis. OK, good
Do you have figures for the soil mineral content? That could help with figuring out sources of feed minerals. No, soil was not investigated. Only pasture grass of pastures 1 and 2 as described in the manuscript due to cost restrictions. OK. You might suggest others take a look at soils in the future.
Contributions of local industrial sources to high Mo make sense, but the research design presented does not address that directly. For this specific site, were excess levels of Cd, Pb and other heavy metals found in the soil air, water, plants, feed or sheep? How about Zn? As described above, this paper is rather supposed to be a detailed case report than a planned research approach. Due to lack of funding no other heavy metals were investigated. However, this is of course a very good idea for further planned research projects addressing those issues in the surrounding of the industrial city of Hamburg. OK
Why are on-farm post mortem examinations and tissue sample collections not allowed? It would seem that producers could be educated to dissect their animals, photograph the results and collect |
|
useful samples for submission. Are vets not allowed to do on-farm post mortem, either, or just farmers? OK. I was asking for my own education, due to my unfamiliarity with Gerrman health rules. Not really part of the review.
In Germany is it forbidden by law to perform on farm PM investigations regardless whether they are performed by the farmer or a veterinarian. All carcasses for PM regardless which animal species need to be send in to specialised labs. Some bigger veterinary practices have their own necropsy room approved by the local veterinary authorities but this is not a standard. Thus, PM investigations are rather expensive and therefor not performed as often as they should. OK. Thanks for that information. |
4
Last comments (not criticism or review):
It looks like you might have a serious environmental load of Mo in that area. A good multiple mineral survey of industrial effluent and fly ash, and new maps of minerals in soil, water, plants, animal tissues and excreta would be helpful in demonstrating and attacking the problem.
Molybdenum is a sneaky environmental element. In the past, American sewage sludge was used as fertilizer. We mix household, animal and factory effluents to some extent. So, even after the more obvious heavy metals (Cd, PB, etc.) were identified as a problem and loads of sludge were monitored for them, the problem with Mo continued. It seems that Mo from lubricants and other products used in maintenance, repair and small-scale manufacturing found its way into pastures and feed crop and caused problems. Naturally high soil Mo also can cause problems for sheep farmers who are trying to be careful about Cu intakes and wind up with black sheep that turn grey.
Author Response
Based on the comments of reviewer 1 the manuscript was changed as follow:
Line 365 ff – changes and amendments are written in italic: Veterinary diagnostic investigations indicate a concomitant molybdenosis, severe secondary Cu deficiency, OWLD (Co deficiency) and Se deficiency in this pedigree sheep flock. Small numbers of investigated animals and samples limit the scope of interferences between the different minerals in this case report, so that the conclusions concerning their interference should be taken with some caution.
Line 367-368 it was added: „and 3 out of 5 animals tested for Se showed a Se deficiency.“
Line 380 ff- words in cursiv were added: Due to the elevated Mo contents in two examined liver samples and the feed samples, a clinical picture of molybdenosis was assumed, and we conclude with caution that this led to secondary Cu deficiency.
